# Assessment of the Macrophage Scavenger Receptor CD163 in Mediating *Glaesserella parasuis* Infection of Host Cells

**DOI:** 10.3390/vetsci10030235

**Published:** 2023-03-21

**Authors:** Xiangwei Deng, Shuilian Li, Ying Zhu, Bo Yu, Jing Zhang, Qianhai Fang, Zhimin Li, Hongbo Chen, Huanhuan Zhou

**Affiliations:** Laboratory of Genetic Breeding, Reproduction and Precision Livestock Farming & Hubei Provincial Center of Technology Innovation for Domestic Animal Breeding, School of Animal Science and Nutritional Engineering, Wuhan Polytechnic University, Wuhan 430023, China; dengxw1999@163.com (X.D.); li2352219017@126.com (S.L.); 18772483460@163.com (Y.Z.); wonderfish@whpu.edu.cn (B.Y.); judyzhang1103@126.com (J.Z.); fangqianhai_98@163.com (Q.F.); ahlizhimin@163.com (Z.L.)

**Keywords:** *Glaesserella parasuis*, CD163, receptor, adhesion, host–pathogen interaction

## Abstract

**Simple Summary:**

The macrophage CD163 surface glycoprotein is a member of the scavenger receptor cysteine-rich (SRCR) family class B. It has been identified as the receptor for hemoglobin–haptoglobin (Hb-Hp) complexes and erythroblasts, and it is the key trigger in host–pathogen interactions. Previous studies have implicated porcine CD163 in macrophage activation delay upon infection with virulent *G. parasuis* strains, while its exact roles in sensing *G. parasuis* infection have not yet been assessed. Here, we investigated the role of CD163 in mediating the adhesion and immune response of *G. parasuis* using in vitro host–pathogen interaction models. We provide evidence that CD163 plays a minor role, unlike those seen in infections with other pathogens, in mediating *G. parasuis* infection.

**Abstract:**

The macrophage CD163 surface glycoprotein is a member of the SRCR family class B, which has been identified as the key trigger in host–pathogen interactions, but its specific roles in sensing *Glaesserella parasuis* (*G. parasuis*) infection are largely unknown. Here, we investigated porcine CD163 in mediating the adhesion and immune response of *G. parasuis* using in vitro host–bacteria interaction models. CD163-overexpressing Chinese hamster ovary K1 cells (CHO-K1) showed obvious subcellular localization in the cytoplasm, especially in the cytomembrane. Although detection using scanning electron microscopy (SEM) confirmed the bacterial adhesion, there was no significant difference in the adhesion of *G. parasuis* to CHO-K1 cells between the presence and absence of CD163. In addition, similar results were observed in 3D4/21 cells. Meanwhile, bindings of *G. parasuis* to nine synthetic peptides, the bacterial binding motifs within SRCR domains of CD163, were weak based on a solid-phase adhesion assay and agglutination assay. Moreover, CD163 had no effect on the expression of *G. parasuis*-induced inflammatory cytokines (IL-6, INF-γ, IL-10, IL-4 and TGF-β) in CHO-K1 cells. In conclusion, these findings indicate that porcine CD163 plays a minor role in sensing *G. parasuis* infection.

## 1. Introduction

*G. parasuis* is a small, nonmotile, pleomorphic rod-shaped and Gram-negative bacterium [1,2]. It is normally colonized in the upper respiratory tract of pigs and develops into a serious systemic inflammatory disease called Glässer’s disease under stress circumstances, which are featured by meningitis, arthritis and polyserositis. In other conditions, it can cause pneumonia and septicemia, resulting in high morbidity or mortality in swine herds (nursing pigs, finishing pigs and sows) [3,4,5]. *G. parasuis* can impair hosts’ lung defense, which probably causes co-infections with other kinds of pathogens, for instance, porcine reproductive and respiratory syndrome virus (PRRSV) [6] and porcine circovirus type 2 virus (PCV2) [7]. Co-infections would negatively affect the body condition, leading to meat production loss in infected pigs. There are drawbacks with the existing prevention strategies. For example, with the usage of antibiotics, more antibiotic resistance genes have been conferred to *G. parasuis* due to the fact of its evolution [8,9]. Moreover, the actual protection effect of some candidate vaccines in pigs is still nonideal [10,11]. An earlier study has shown that 10% of all experimentally infected pigs were fully resistant, suggesting that there are differences in susceptibility to *G. parasuis* infection in pigs. These data support the hypothesis that sires influence their offspring’s response to infection [12]. Thus, identifying the genetic factors associated with the differences in susceptibility to *G. parasuis* infection is a promising way to selectively breed pigs that are more resistant to Glässer’s disease. However, the mechanisms of hosts’ genetic control of the infection are still unclear.

Following colonization in the lower respiratory tract, the lung is the primary site for the systemic infection of *G. parasuis* [13], wherein the responses of porcine alveolar macrophages (PAMs) will act as the first line of defense in hosts. Previous research has demonstrated that a high level of CD163 on the surface of PAMs and, subsequently, the large amounts of soluble CD163 (sCD163) in serum were a striking observation in susceptible piglets infected with virulent strains of *G. parasuis* [14], implying the potential role of CD163 in the development of Glässer’s disease. However, the exact mechanism is largely unknown. CD163 is highly expressed on a majority of the subpopulations of resident tissue macrophages, acting as a cell-surface glycoprotein receptor [15]. It is composed of nine SRCR domains in its extracellular peptide. So far, a collection of studies have supported the idea that CD163 is a binding receptor for some bacteria and viruses during the process of infection [16,17]. In pigs, CD163 acts as a key receptor on host cells for African swine fever virus (ASFV) [18]. CD163 knockout pigs can survive when exposed to highly pathogenic PRRSV [17,19]. Therefore, we hypothesized that porcine *CD163* might also be an important candidate gene (or receptor) affecting pig susceptibility to *G. parasuis*, while its role in sensing *G. parasuis* infection has not yet been assessed. 

In the present study, we systematically assessed the exact role of porcine CD163 in mediating *G. parasuis* infection using bacterial adhesion, solid-phase adhesion, agglutination assay and real-time quantitative PCR (Q-PCR). Our results provide a reference for the better understanding of the interactions between *G. parasuis* and pigs.

## 2. Materials and Methods

### 2.1. Cell Culture

The CHO-K1 cells were cultured in complete growth media containing Dulbecco’s Modified Eagle Medium/Nutrient Mixture F-12 (Gibco, Grand Island, NY, USA) supplemented with 10% fetal bovine serum (Gibco, USA) at 37 °C under 5% CO_2_ in a humidified cell incubator. The 3D4/21 cells were cultured in the same way, except that the complete growth media consisted of Roswell Park Memorial Institute 1640 (Gibco, USA) supplemented with 10% fetal bovine serum (Gibco, USA).

### 2.2. Vector Construction, Transient Transfection and Stable Overexpression

The porcine CD163 coding sequence was amplified by the forward primer 5′-aattGGATCCATGGACAAACTCAGAATGGTGC-3′ and the reverse primer 5′-aattCTCGAGTCATTGTACTTCAGAGTGGTCTCC-3′ inserted into pcDNA3.1(+) linearized by *Xho*I and *Bam*HI (NEB, Ipswich, MA, USA) and then verified by Sanger sequencing (Thermo Fisher Scientific, Waltham, MA, USA). The endotoxin-free plasmids were extracted and used for the subsequent experiments. The day before transient transfection, the CHO-K1 cells or 3D4/21 cells were plated in complete growth media without antibiotics. According to the manufacturer’s instructions for Lipofectamine™ 3000 Transfection Reagent (Invitrogen, Waltham, MA, USA), the CHO-K1 cells or 3D4/21 cells were transfected with pcDNA3.1-CD163 (treatment group) or pcDNA3.1 (control group). In particular, to select the CHO-K1 cells stably expressing porcine CD163, 1 mg/mL G418 (Solarbio^®^, Beijing, China) was added to the complete growth media after 48 h in both the treatment group and control group, until almost all cells in the control group died. Then, these cells in the treatment group were expanded and screened by Sanger sequencing for further assays.

### 2.3. Immunofluorescence

The cells were cultured in 24-well plates, rinsed twice with phosphate buffer solution (PBS) and fixed with ice-cold Immunostaining Fix Solution (Beyotime Biotechnology, Shanghai, China) for 20 min, followed by washing with PBS two times. Then, the cells were incubated in ice-cold Immunostaining Permeabilization Buffer (Beyotime, China) for 10 min and washed twice with PBS. The cells were then incubated in Immunostaining Blocking Buffer (Beyotime Biotechnology, China) at room temperature for 60 min, followed by washing with PBS three times. Next, the cells were incubated with the following primary antibody, Mouse anti Pig CD163 Monoclonal Antibody (Bio-Rad, Berkeley, CA, USA, MCA2311GA), at 4 °C overnight, and they were washed twice with PBS. Next, the cells were incubated with Goat anti Mouse IgG (H/L) (DyLight^®^488, Bio-Rad, USA) for 1 h in the dark at room temperature, following by washing with PBS three times. The cells were then stained with 4′,6-diamidino-2-phenylindole (DAPI) in the dark for 5 min and then rinsed three times with PBS. The images were visualized using an Olympus/IX73 TH4-200 system (Olympus, Tokyo, Japan) or Olympus/FV10i-O system (Olympus, Japan). Three independent experiments were performed for immunofluorescence detection. 

### 2.4. Bacterial Strain Culture 

In this study, a highly virulent strain of serovar 5 (*G. parasuis* SH0165 strain) was used. *G. parasuis* was cultured at 37 °C on trypticase soy agar (TSA) (Difco^TM^, BD, Radnor Township, PA, USA) and in trypticase soy broth (TSB) (Difco^TM^, BD, USA) and supplemented with 10% fetal bovine serum (Gibco, USA) and 0.01% nicotinamide adenine dinucleotide (Sigma-Aldrich^®^, Taufkirchen, Germany). A single *G. parasuis* colony growing on TSA solid medium was selected to be cultured in TSB liquid medium overnight in a constant temperature shaker at 37 °C with 225 rpm/min until the optical density (OD) of the measured culture reached approximately 0.6 to 0.7 at 645 nm.

### 2.5. Adhesion Assays 

An adhesion assay was performed to quantify the total cell-associated (surface-adhered plus intracellular) bacteria. The cells were plated in 24-well plates (2 × 10^5^ cells/well) in triplicate and infected with *G. parasuis* when developing into confluent monolayers. To allow bacterial adhesion, the cocultures were incubated for 6 h at 37 °C under 5% CO_2_ in a humidified cell incubator. Thereafter, the cells were rinsed seven times with PBS and incubated with 200 µL of 0.25% trypsin (Gibco, USA) at 37 °C. Next, the cells were lysed with 800 µL of ice-cold double-distilled water and disrupted by scraping them with cell scrapers. Then, the dilutions of the cell lysate were seeded onto TSA solid medium and incubated for 48 h at 37 °C for colony-forming unit (CFU) determination. The level of adhesion was calculated as the proportion of total cell-associated bacteria to the total inoculated bacteria. To verify the results, an adhesion assay was independently performed in triplicate by two proficient experimenters. 

### 2.6. Adhesion Study by Scanning Electron Microscope (SEM) 

The *G. parasuis* adhesion was also determined by SEM. Confluent monolayers of CHO-K1 cells, cultured on a 13 mm Thermanox coverslip in a 24-well plate, were infected and incubated as described above. After washing with PBS three times, the monolayers were fixed with precooled 2.5% glutaraldehyde for 1 h at room temperature. Then, the samples were washed three times with PBS and fixed again using 2% osmium tetroxide in deionized water for 15 min at room temperature. Next, these samples were sequentially dehydrated in ethanol solutions of different gradient concentrations, then dried in a critical point dryer, coated with gold–palladium by an ion sputtering coating method and, finally, viewed with a JEOL microscope. Three independent experiments were performed for the adhesion study via SEM.

### 2.7. Real-Time Quantitative PCR (Q-PCR) 

The total RNA of the cells was isolated from cell samples using Trizol (TIANGEN, Beijing, China) and reverse transcribed to cDNA with a PrimeScript™ 1st Strand cDNA Synthesis Kit (TaKaRa, Shiga, Japan) according to the manufacturer’s protocol. The Q-PCR was conducted with TB Green^®^ Premix Ex Taq^TM^ (TaKaRa, Japan) in 96-well plates using a 7500 real-time PCR detection system (ABI, Waltham, MA, USA). All reactions were performed in triplicate. The relative expression was calculated using the 2^−∆∆Ct^ method [20], with β-actin as the internal control. The primers used for the Q-PCR are shown in Appendix A.

### 2.8. Solid-Phase Adhesion Assay 

The solid-phase binding of the bacteria to the synthetic peptide was conducted as described previously [21,22]. In brief, the synthetic peptide (40 μg/mL) was incubated in microtiter plates overnight at 4 °C after dilution in coating buffer (100 mM sodium carbonate, pH 9.6) and then washed twice with the relevant buffers. The samples were added to the *G. parasuis* suspension (100 μL/well, 5 × 10^8^ bacteria/mL), co-incubated for 2 h at 37 °C and washed twice. To label the *G. parasuis* adhering to the synthetic peptide, SYTO™ 9 Green Fluorescent Nucleic Acid Stain (Invitrogen, USA) was added at 100 μL/well and incubated for half an hour in the dark at 37 °C and then washed three times and measured in a SpectraMax M2 microplate reader (Molecular Devices, San Jose, CA, USA) at 485 nm absorption and 535 nm emission. The solid-phase adhesion assay was independently repeated 3 times.

### 2.9. Agglutination Assay

An agglutination assay was conducted as described previously [21,22]. Briefly, 100 μL of the synthetic peptide solution was mixed with 100 μL of the *G. parasuis* suspension (5 × 10^8^ bacteria/mL) in a 96-well microtiter plate, making the final concentration of the synthetic peptide solution 100 μg/mL. Then, they were co-incubated for 2 h at 37 °C. To visualize the agglutination process, a turbidimetric analysis of the suspension of the co-incubator was carried out with a SpectraMax M2 microplate reader (Molecular Devices, USA). The optical density of the *G. parasuis* suspension was measured at 10 min intervals for a total of 180 min at 645 nm. These experiments were repeated at least three times. This experiment was repeated at least three times.

### 2.10. Statistical Analysis 

All data are shown as the mean ± standard deviation (SD). The Student’s *t*-test was used to analyze the statistical significance. Values of *p* < 0.05 were considered statistically significant.

## 3. Results

### 3.1. Overexpression of Porcine CD163 in CHO-K1 Cells

To study the role of CD163 in sensing *G. parasuis* infection, in vitro cell models overexpressing the porcine *CD163* gene were constructed. First, plasmid-expressing CD163 was transiently transfected into nonpermissive CHO-K1 cells. After transfection, a significant overexpression of *CD163* was detected by Q-PCR at the mRNA level (Figure 1A), and the overexpression of the CD163 protein was also detected by immunofluorescence (Figure 1B). Furthermore, CHO-K1 cells stably expressing CD163 were also developed to improve the efficiency of the cell transfection. As shown in Figure 1C, the protein was localized in the cytoplasm, especially on the membrane, indicating that porcine CD163 was successfully expressed in the CHO-K1 cells.

### 3.2. The Role of CD163 in Mediating the Adhesion of G. parasuis to CHO-K1 Cells

To investigate whether CD163 mediates host susceptibility to a virulent *G. parasuis* strain, the adhesion of cell-associated bacteria in *G. parasuis*-infected CHO-K1 cells was analyzed by an adhesion assay and SEM. As shown in Figure 2A, the bacteria could adhere to the CHO-K1 cells, but there was no significant difference in the number of adhered bacteria per cell between the CHO-K1^CD163^ cells (1.7 ± 1.66) and nonpermissive cells (2.96 ± 1.28, equivalent to 2.96% of adhesion). Consistently, the number of adhered bacteria per CHO-K1^CD163′^ cell was slightly higher (10.18 ± 1.08), but there was also no detectable difference compared with the control group (8.47 ± 3.98, Figure 2B). As shown in Figure 2C, the SEM results clearly show that *G. parasuis* could adhere to the surface of CHO-K1 cells, but the presence or absence of CD163 expression in the cells, demonstrated by the adhesion assay, did not significantly change the bacteria’s adhesion.

### 3.3. Effect of CD163 on G. parasuis Adhesion to 3D4/21 Cells 

Since PAMs are the first effector cells exposed to pathogens in the process of host resistance to *G. parasuis* [23,24], 3D4/21 cells, the PAM cell line, were used as the other in vitro model of *G. parasuis* infection. As shown in Figure 3A, the mRNA expression of porcine *CD163* was significantly upregulated in 3D4/21^CD163^ cells, and the protein expression level was also upregulated (Figure 3C). However, the expression of CD163 was incapable of affecting the number of *G. parasuis* adhered to the 3D4/21 cells (Figure 3B). 

### 3.4. Bindings of G. parasuis to the Extracellular SRCR Domains of CD163

Motifs of GRVEVxxxxxW within porcine CD163 scavenger domains have been proved to be able to mediate other bacterial bindings [16]. Here, we intended to further determine whether the corresponding motifs also mediated *G. parasuis* binding. Hence, 9 out of 11 amino acid peptides from porcine CD163 extracellular scavenger domains (CD163p1-9), representing the GRVEVxxxxxW motifs, were artificially synthesized. Next, their binding to *G. parasuis* was tested via a solid-phase adhesion assay. As shown in Figure 4A, no peptide segments displayed a significant binding to *G. parasuis* when compared with the controls. Moreover, a turbidimetric analysis was also carried out to evaluate their agglutination on the bacteria. Similar to the solid-phase adhesion assay, there was no observed agglutination of *G. parasuis* for all peptide segments, and almost all were the same as the negative control group (Figure 4B).

### 3.5. Recognition of G. parasuis by CD163 in Triggering Inflammatory Factors 

It is well known that, normally, the expression of cellular inflammatory cytokines changes in order to engage in immune system response when the host is infected by pathogens. In *G. parasuis*-infected CHO-K1^CD163′^ cells and CHO-K1 cells, the mRNA level of three anti-inflammatory factors, IL-10, IL-4 and TGF-β, and two pro-inflammatory factors, IL-6 and INF-γ, was detected by Q-PCR. Consistent with the above findings, the overexpression of CD163 did not alter the expression of these inflammatory cytokines induced by *G. parasuis* infection (Figure 5). 

## 4. Discussion

*G. parasuis* infection can cause Glässer’s disease, usually as a result of co-infection with secondary pathogens such as PRRSV [6], resulting in a sharp increase in morbidity and mortality in swine populations. However, the molecular mechanism underlying the pathogenesis of *G. parasuis* remains poorly understood at present, and the associated resistance or susceptibility genes in pigs remain unclear. Growing evidence supports that CD163 mediates the susceptibility of other bacteria and viruses to hosts and, more importantly, an increased CD163 expression is a dramatic feature in experimentally infected pigs with *G. parasuis* [25,26]. 

To investigate the exact role of CD163 in sensing *G. parasuis*, the adhesion of *G. parasuis* to model cells was first tested, and the results clearly show that the bacterium was able to adhere to CHO-K1 cells in vitro (up to 8.47%). Similar to our results, the bacterial adhesion value reached 7.44% when testing highly virulent and avirulent serum strains using PK-15 cells [27]. Whereas PBMEC/C1-2 cells, a porcine brain microvascular endothelial line, had lower adhesion values (<0.1%) when tested with different *G. parasuis* serotypes [5]. In addition, the adhesion value of the 3D4/21 cells associated with the adhesion of *G. parasuis* was approximately 1% in our study, which is, overall, higher than that in previous studies [28,29]. In brief, our results confirm that CHO-K1 cells, which were as susceptible to *G. parasuis* as 3D4/21 cells, are a useful in vitro infection model. 

Moreover, the number of total cell-associated *G. parasuis* did not change in either the CHO-K1 cells or 3D4/21 cells when CD163 was overexpressed, which is in agreement with our SEM results, indicating the small role of CD163 in sensing *G. parasuis* infection. Contrary to our results, a study has shown that human CD163 acts as a macrophage surface sensor for the recognition of both intact Gram-negative and Gram-positive bacteria, including *S. mutans*, *S. aureus* and *E. coli* [16]. Research on *S. mutans* has mapped their bacterial binding site to the CD163p2 and CD163p3 peptide motifs in the second and third SRCR domains of human CD163, respectively [16]. In pigs, porcine CD163 has also been proved to be the key receptor of ASFV [18] and PRRSV [19]. When lacking the CD163 SRCR5 domain, porcine alveolar macrophages isolated from genome-edited pigs were fully resistant to PRRSV; thus, their normal biological function is just as important [30]. However, none of the corresponding peptide motifs within porcine CD163 were capable of binding *G. parasuis*, indicating different pathogen–host interaction mechanisms among different pathogens. We should not totally exclude certain roles of other peptides within CD163 in mediating infection, especially the interaction between CD163 and the virulence factors of *G. parasuis*, although based on previous reports [16,18,19,21,22] and our investigation, we presume that this possibility is very low.

As an innate immune sensor and inducer of local inflammation, porcine CD163 promotes inflammatory cytokine production induced by Gram-positive *Actinobacillus pleuropneumoniae* (APP) in monocytes/macrophages [31]. In addition, studies in humans support that cytokine production can be triggered by signals responding to bacterial recognition by CD163 [16]. Conversely, we found that it was unable to affect the expression of cellular inflammatory cytokines following *G. parasuis* infection, which further indicates the minor role of CD163 in recognizing this bacterium.

Nonvirulent *G. parasuis* strains can be efficiently phagocytosed by PAMs, while virulent strains can resist phagocytosis [23]. After the first day of in vivo infection, nonvirulent strains could induce increased levels of CD163 on the surfaces of PAMs, but virulent strains, contrarily, led to a reduced expression of CD163, suggesting that an increased expression of CD163 may indicate that nonvirulent strains are susceptible to phagocytosis and, thereafter, these strains are easily killed by PAMs [14,23]. With the processing of infection, the results observed at day 1 switched to a strong elevation of surface CD163 by the virulent strains and, subsequently, increased levels of sCD163, which has been recognized as an evident biomarker of sepsis, comorbidity, mortality and macrophage activation syndrome [32] in serum (days 3–4) [14]. Taking into account our herein investigation, the elevated expression of surface CD163 does not mean that it would act as a sensor for *G. parasuis*. In agreement with this, there was no specific receptor for the phagocytosis of *G. parasuis* shown by the competition assays [23]. The conversed inductions of surface CD163 by virulent strains could also reflect an early inhibition of the inflammatory response, leading to the failure of bacterial clearance during the process of systemic infection. On the other hand, high levels of proteolytically shed CD163 in serum may indicate a shutdown mechanism for excessive inflammatory cytokine production at the later stage of infection, as speculated by Fabriek et al. (2009) [16], while investigating the synergistic roles of CD163 with other receptor(s), such as Siglec1 and ASFV infection [18], in the process of *G. parasuis* infection could be a way to uncover the mediating mechanism of CD163. The work on discovering resistant or susceptible candidates of *G. parasuis* infection never stops, and genome wide screening based on CRISPR/Cas9 technology has started in our lab.

## 5. Conclusions

In summary, on the basis of our results, it can be suggested that porcine CD163 may play a minor role, at least with it alone, in mediating *G. parasuis* infection.

## Figures and Tables

**Figure 1 vetsci-10-00235-f001:**
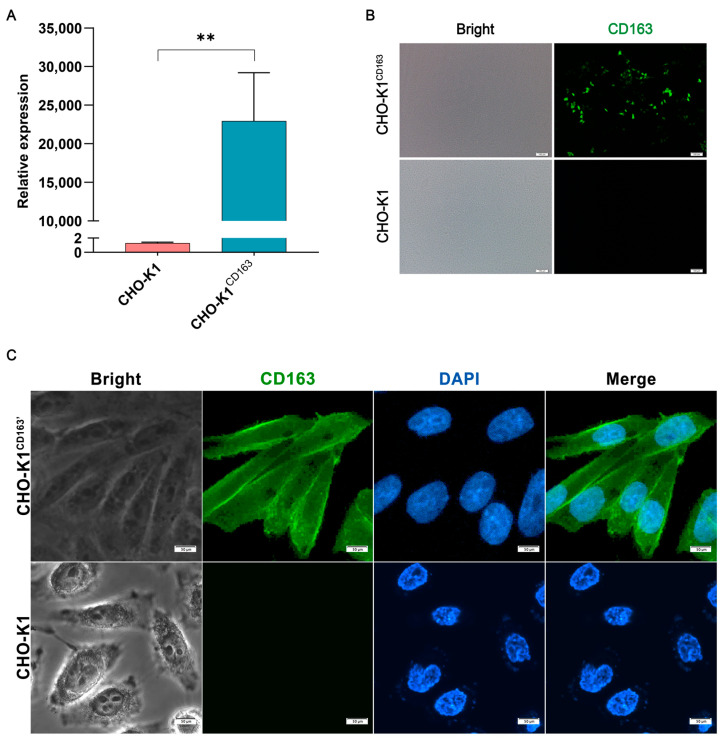
Construction of the porcine CD163-expressing CHO-K1 cell models. (**A**) The mRNA expression of porcine CD163. (**B**,**C**) The protein expression pattern of CD163 by immunofluorescence. CHO-K1^CD163^: porcine CD163 overexpressed in CHO-K1 cells by transient transfection. CHO-K1^CD163′^: CHO-K1 cells stably overexpressing porcine CD163. The proteins are visualized with an anti-CD163 antibody (green), and the nuclei are stained with DAPI (blue). ** *p* < 0.01.

**Figure 2 vetsci-10-00235-f002:**
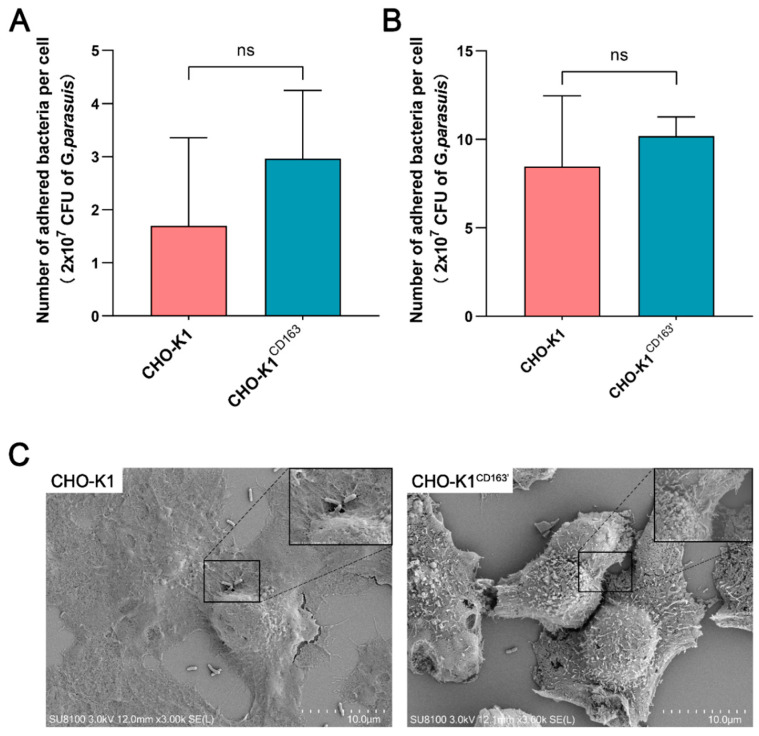
The adhesion of *G. parasuis* to CHO-K1 cells with or without CD163 expression. (**A**,**B**) The average number of adhered *G. parasuis* per cell at 6 h incubation. The bacterial inoculum tested was 2 × 10^7^ CFU. (**C**) SEM micrograph showing the attachment of *G. parasuis* to the surface of CHO-K1 cells or CHO-K1^CD163′^ cells after incubation for 6 h. CHO-K1^CD163^: porcine CD163 overexpressed in CHO-K1 cells by transient transfection. CHO-K1^CD163′^: CHO-K1 cells stably overexpressing porcine CD163. ns: Nonsignificant.

**Figure 3 vetsci-10-00235-f003:**
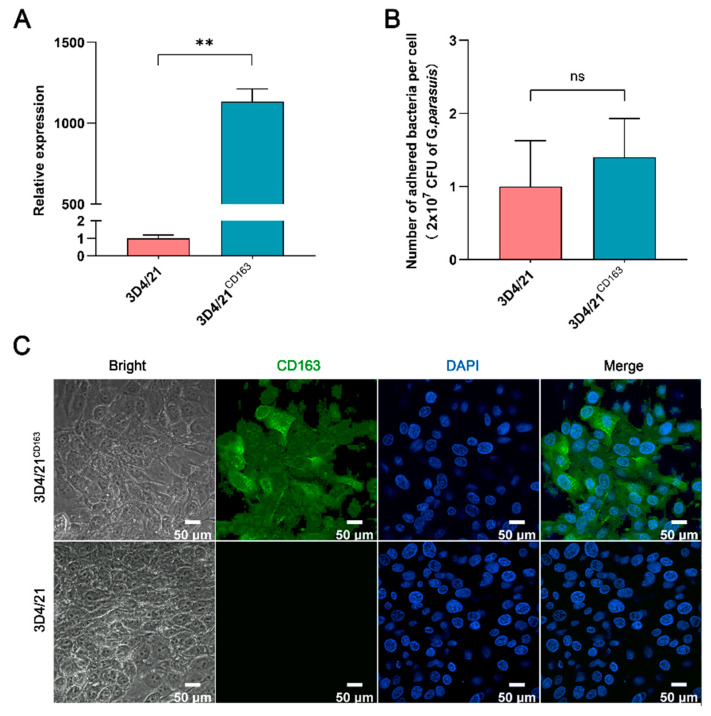
Adhesion analysis of *G. parasuis* to 3D4/21 cells with or without CD163 expression. (**A**) A Q-PCR assay of the *CD163* mRNA expression in 3D4/21 cells. 3D4/21^CD163^: porcine CD163 gene overexpressed in 3D4/21 cells by transient transfection. (**B**) A statistical analysis of *G. parasuis* adhesion to 3D4/21cells. ** *p* < 0.01; ns: Nonsignificant. (**C**) Immunofluorescence detection of the CD163 protein expression in 3D4/21 cells. The proteins are visualized with an anti-CD163 antibody (green), and the nuclei are stained with DAPI (blue).

**Figure 4 vetsci-10-00235-f004:**
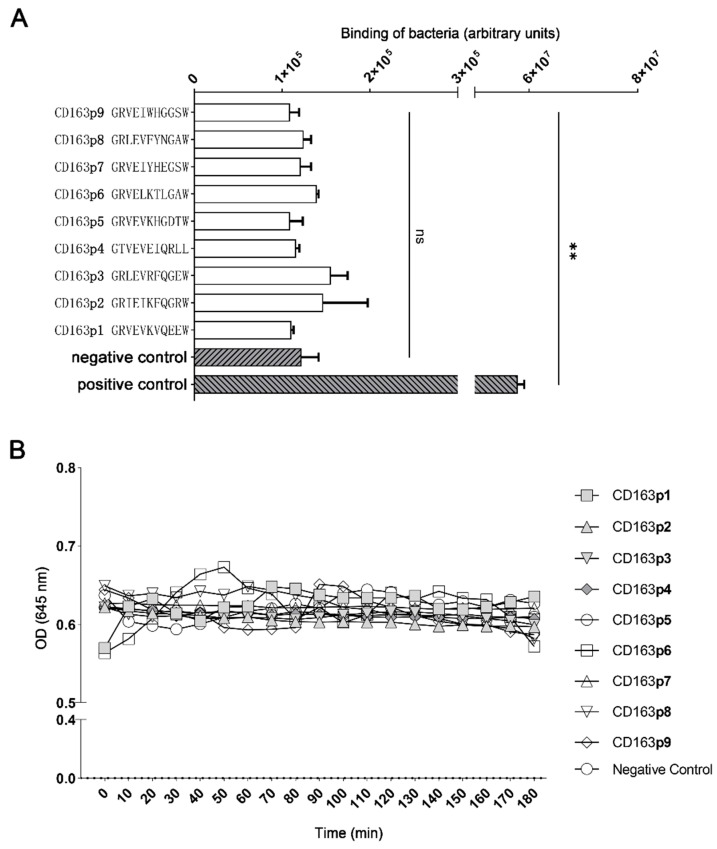
Binding of *G. parasuis* to 11-mer extracellular peptide motifs within SRCR domains (1–9) of porcine CD163. (**A**) A solid-phase adhesion assay of *G. parasuis* binding to 9 corresponding motifs within SRCR domains of porcine CD163. Negative control: scramble peptide; positive control: *G. parasuis* labeled with SYTO™ 9. (**B**) The turbidimetric agglutination of *G. parasuis* with the above peptides. ** *p* < 0.01. ns: Nonsignificant.

**Figure 5 vetsci-10-00235-f005:**
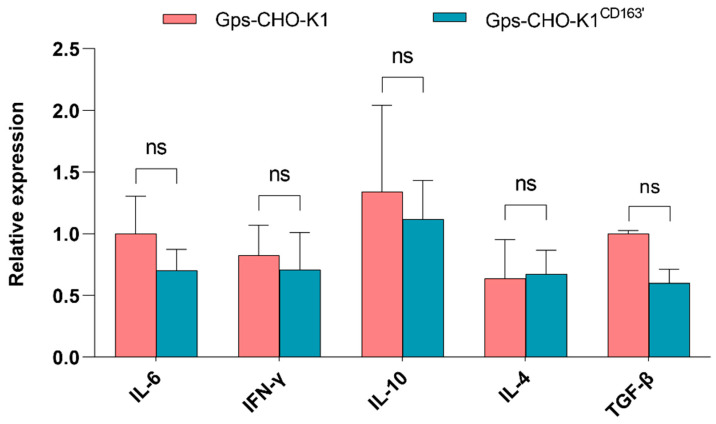
mRNA expression of five cellular inflammatory factors. The relative mRNA expression level of *IL6*, *IFN-γ*, *IL-10*, *IL-4* and *TGF-β* induced by *G. parasuis* in CHO-K1 cells with or without CD163 expression was evaluated via Q-PCR. ns: Nonsignificant.

## Data Availability

The data that support the findings of this study are available from the corresponding author upon reasonable request.

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
