# Peer review of "Assessment of the Macrophage Scavenger Receptor CD163 in Mediating Glaesserella parasuis Infection of Host Cells"

_vetsci, 2023, doi:10.3390/vetsci10030235_

Round 1

Reviewer 1 Report

It is well known that CD163 is a key receptor for some pathogens in the pig, such as PRRSV and ASFV, and it has been proved this protein recognizes both G+ and G- bacteria in human. In this manuscript, the authors found CD163 may not be the receptor of Glaesserella parasuis infection. The results, based on adhesion & agglutination assay, SRCRs- Glaesserella parasuis interaction evaluation, and CKs assessment. This study offered some new information for further study the infection of the Glaesserella parasuis. 

The result showed that the CD163 is not sensitive for the infection of Glaesserella parasuis, which is inconsistent with the meaning of the title.

For each research, the number of repeated experiments or samples should be marked.

For Figure 2, how to count the adhered bacteria in the electron microscope picture.

For figure 4, the author used only peptide segments to do the adhesion experiment. Why not consider using the whole gene to do this experiment.

Author Response

Response to Reviewer 1 Comments
Note: All changes according to reviewer 1 comments are marked in blue words and highlighted in yellow in the revised manuscript.
It is well known that CD163 is a key receptor for some pathogens in the pig, such as PRRSV and ASFV, and it has been proved this protein recognizes both G+ and G- bacteria in human. In thismanuscript, the authors found CD163 may not be the receptor of Glaesserella parasuis infection.
The results, based on adhesion & agglutination assay, SRCRs-Glaesserella parasuis interaction evaluation, and CKs assessment. This study offered some new information for further study the
infection of the Glaesserella parasuis.
Response: Thank you very much for your overall comments on our work. As far as we know, this is the first work on the interplay of porcine CD163 and Glaesserella parasuis despite the preliminary job. Hopefully, this investigation will help to increase our understanding of the infection or host response.
Point 1: The result showed that the CD163 is not sensitive for the infection of Glaesserella parasuis, which is inconsistent with the meaning of the title.
Response 1: Yes, we agree with your comment. The title has been modified for “Assessment of the macrophage scavenger receptor CD163 in mediating Glaesserella parasuis infection of host
cells”. Please see line 2 in the revised manuscript.
Point 2: For each research, the number of repeated experiments or samples should be marked.
Response 2: Sure, we should mark the replicates or the repeated times of independent experiment.
All of the info. have been added in the M&M sections. Please see lines 116-117, 129-130, 137-139, 149-150, 156, 168-169, 178-179 in the revised manuscript.
Point 3: For Figure 2, how to count the adhered bacteria in the electron microscope picture.
Response 3: The average adhered bacteria on cells were “counted” via the adhesion assay (M&M, section 2.5.), which is a canonical method in quantitative determination of bacterial adhesion. The main object of SEM investigation was focused on microscopic analysis, in which the interaction between G. parasuis and host cells could be intuitively presented. In case of misunderstan, we modified the description in the text again. Please see lines 29-30, 141, 203-206.
Point 4: For figure 4, the author used only peptide segments to do the adhesion experiment. Why not consider using the whole gene to do this experiment.
Response 4: This is a very good question! In fact, we had indirectly assessed the G. parasuis adhesion of host cells using the adhesion assay, at the “whole gene” level, in a CD163 protein overexpression way. Because CHO-K1 cells express little or no endogenous CD163 (Figure 1, A-C), the CD163-overexpressing CHO-K1 cells Vs. CHO-K1 cells can be a good model to achieve such assessment. It has been proved that the SRCRs of CD163 play crucial roles in the process of
interplay between this protein and its targets, including viruses, bacteria, and hemoglobin-haptoglobin (Hb-Hp) complexes (Bikker et al., 2002; 2004; 2007; Fabriek et al., 2009). We might not totally exclude roles of other peptides in CD163 in mediating G. parasuis infection, especially the interaction between CD163 and the virulence factors of G. parasuis. While based on previous investigations and our herein adhesion assay, we presume that the possibility will be very low. In addition, we do have the direct interplay assay method to evaluate CD163- G. parasuis adhesion, in which we should get the pure CD163 protein for the first of all, and then detect the interaction
between CD163 and G. parasuis lysates. The key trouble is we don’t have the instrument in hand right now, such as the The Octet® N1 System. Our team will buy one in 2023 (single channel),
hopefully we will figure out this in the future. We added the related discussion in the section 4, please see lines 300-304 in the revised
manuscript.
Again, we appreciate your comments and consideration of our work!

Reviewer 2 Report

This manuscript is useful in understanding the role, albeit minor role, of CD163 in G. parasuis infection. It is a useful step in fully elucidating the mechanisms of host-pathogen interactions in this system. The manuscript is overall scientifically sound, in my opinion. There are multiple grammatical changes that must be made before publication, however. My comments are outlined below.

Simple summary, line 22: this sentence phrasing sounds a bit off; I would re-phrase to something more like “… CD163 plays a minor role, unlike those seen in infections with other pathogens, in sensing…”

Line 47: “… which probably causing co-infections…” should be changed to “… which probably causes co-infections…”

Line 51: change to something like “… with the usage of antibiotics, more antibiotic resistance genes have been conferred to G. parasuis…

Line 53: maybe be a bit more specific here when citing this earlier study. Naturally infected pigs in the wild? Experimentally infected pigs? Also I would cite this sentence with the appropriate publication.

Line 60: I don’t think “deposition” is the right word for what you are trying to say here. Try something like “colonization”, “binding”, or “infection”.

Line 66: It may not be appropriate to say that CD163 causes Glasser’s disease; it isn’t the pathogen.

Line 226: change “… 11-amina acid…” to “… 11-amino acid…”

Line 228: You don’t need the “Nevertheless” here.

Figure 4B: could you change range of the y-axis of the figure? The range from 0.0 to 1.0 makes discerning any differences between the lines on the graph impossible.

Line 254: Edit “Growing evidences support…” to “Growing evidence supports…”

Line 257: I would consider re-phrasing this; it is a bit unclear what you mean by CD163 being sensitive to G. parasuis infection, as CD163 itself can’t be infected.

Line 296: Replace “increasement” with “increased expression” or “increasing expression”

Author Response

Response to Reviewer 2 Comments
Note: All changes according to reviewer 2 comments are marked in red words and highlighted in yellow in the revised manuscript.
Point 1: line 22: this sentence phrasing sounds a bit off; I would re-phrase to something more like “... CD163 plays a minor role, unlike those seen in infections with other pathogens, in sensing...”
Response 1: Yes. We have modified according to the suggestion. Please see line 22 in the revised manuscript.
Point 2: Line 47: “... which probably causing co-infections...” should be changed to “... which probably causes co-infections...”
Response 2: Yes. Please see line 47 in the revised manuscript.
Point 3: Line 51: change to something like “... with the usage of antibiotics, more antibiotic resistance genes have been conferred to G. parasuis... “
Response 3: Yes. Please see line 51-52 in the revised manuscript.
Point 4: Line 53: maybe be a bit more specific here when citing this earlier study. Naturally infected pigs in the wild? Experimentally infected pigs? Also I would cite this sentence with the appropriate publication.
Response 4: Yes. We have modified according to the suggestion. Please see line 54 in the revised manuscript. And, we are very pleased to see that you are interested in this reference!
Point 5: Line 60: I don’t think “deposition” is the right word for what you are trying to say here. Try something like “colonization”, “binding”, or “infection”.
Response 5: Thank you! Expression shadow of Chinese! We have modified according to the suggestion. Please see line 60 in the revised manuscript.
Point 6: Line 66: It may not be appropriate to say that CD163 causes Glasser’s disease; it isn’t the pathogen.
Response 6: Yes. Please see line 66 in the revised manuscript.
Point 7: Line 226: change “... 11-amina acid...” to “... 11-amino acid...”
Response 7: Thank you! We have corrected the writing. Please see line 239 in the revised manuscript.
Point 8: Line 228: You don’t need the “Nevertheless” here.
Response 8: Yes. Please see line 241-242 in the revised manuscript.
Point 9: Figure 4B: could you change range of the y-axis of the figure? The range from 0.0 to 1.0 makes discerning any differences between the lines on the graph impossible.
Response 9: Good suggestion! We have corrected according to the suggestion. Please see Figure 4B in the revised manuscript. It looks much better.
Point 10: Line 254: Edit “Growing evidences support...” to “Growing evidence supports...”
Response 10: Yes. Please see line 268 in the revised manuscript.
Point 11: Line 257: I would consider re-phrasing this; it is a bit unclear what you mean by CD163 being sensitive to G. parasuis infection, as CD163 itself can’t be infected.
Response 11: Thank you. We have corrected according to the suggestion. Please see line 270-271 in the revised manuscript.
Point 12: Line 296: Replace “increasement” with “increased expression” or “increasing expression”
Response 12: Yes. Please see line 315 in the revised manuscript.
We thank you very much for the comments that help to improve the writing much more!